# ConRad: Image Constrained Radiance Fields for 3D Generation from a Single Image

**Senthil Purushwalkam**[*]
Salesforce AI Research
spurushwalkam@salesforce.com

**Nikhil Naik**
Salesforce AI Research
nnaik@salesforce.com

## Abstract

We present a novel method for reconstructing 3D objects from a single RGB image. Our method leverages the latest image generation models to infer the hidden 3D structure while remaining faithful to the input image. While existing methods[1, 2] obtain impressive results in generating 3D models from text prompts, they do not provide an easy approach for conditioning on input RGB data. Naïve extensions of these methods often lead to improper alignment in appearance between the input image and the 3D reconstructions. We address these challenges by introducing *Image Constrained Radiance Fields (ConRad)*, a novel variant of neural radiance fields. ConRad is an efficient 3D representation that explicitly captures the appearance of an input image in one viewpoint. We propose a training algorithm that leverages the single RGB image in conjunction with pretrained Diffusion Models to optimize the parameters of a ConRad representation. Extensive experiments show that ConRad representations can simplify preservation of image details while producing a realistic 3D reconstruction. Compared to existing state-of-the-art baselines, we show that our 3D reconstructions remain more faithful to the input and produce more consistent 3D models while demonstrating significantly improved quantitative performance on a ShapeNet object benchmark.

## 1 Introduction

Humans posses the ability to accurately infer the full 3D structure of an object even after observing just one viewpoint. Since the RGB and depth observed for one viewpoint does not provide sufficient information, we have to rely on our past experiences to make intelligent inferences about a realistic reconstruction of the full 3D structure. This ability helps us navigate and interact with the 3D world around us seamlessly. Capturing a similar capability of generating the full 3D structure from a single image has been a long-standing problem in Computer Vision. Such systems could facilitate advances in robotic manipulation and navigation systems, and has applications in video games, augmented reality and e-commerce. Despite the importance of this capability, success on solving this problem has been limited.

This lack of success can partly be attributed to the lack of useful large scale data. While we have made progress on collecting large scale image datasets, the scale of 3D object or multi-view image datasets has been severely limited. The availability of billions of RGB images and captions [3] has facilitated significant advances in semantic understanding of images [4–6]. Over the last few years in particular, approaches for generative modeling of the image manifold have demonstrated capabilities including controllable generation of highly realistic images [7, 8], inpainting of occluded images [9] and realistic image editing [10–12]. These success stories demonstrate the ability of the 2D generative models to capture prior knowledge about the visual world. In light of this evidence,

---

[*]Project webpage: https://www.senthilpurushwalkam.com/publication/conrad/

37th Conference on Neural Information Processing Systems (NeurIPS 2023).

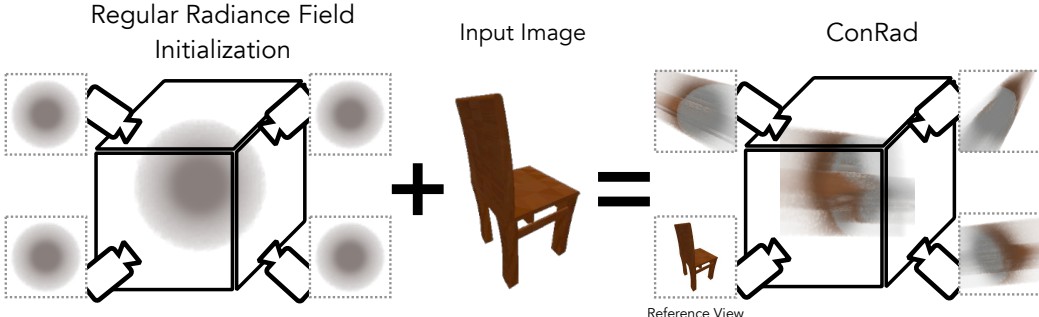

**Figure 1: ConRad representation:** We propose a novel radiance field,ConRad, which allows conditioning on a single image. In contrast to regular radiance fields, our approach can accurately model the input image in one reference viewpoint of the radiance field without any training. Utilizing ConRad representations simplifies the optimization of a radiance field for generating a 3D model from a single image.

we ask the question—can this 2D prior knowledge be leveraged to infer the hidden 3D structure of objects?

DreamFusion [1] generates 3D models based on an input text prompt, using the distribution captured by a pretrained diffusion model to update the parameters of a neural radiance field (NERF) [13]. While DreamFusion demonstrates impressive capabilities, generating arbitrary instances of objects from text prompts severely limits its applications. In recent concurrent work, RealFusion [14] and NeuralLift360 [15] extend DreamFusion to accommodate image input. These methods propose to optimize additional objectives to reconstruct the given input image in one reference viewpoint of the NERF while still relying on a diffusion model to generate a realistic reconstruction in novel viewpoints.

In this work, we propose a novel method for generating a 3D object from a single input image. Instead of designing additional objectives to reconstruct the input image, we propose to rethink the underlying 3D representation. We introduce Image Constrained Radiance Fields (ConRad), a novel variant of NERFs that can explicitly capture an input image without any training. Given an input image and a chosen arbitrary reference camera pose, ConRad utilizes multiple constraints to incorporate the appearance of the object and the estimated foreground mask into the color and density fields respectively. These constraints ensure that the rendering of the radiance field from the chosen viewpoint exactly matches the input image while allowing the remaining viewpoints to be optimized.

The proposed ConRad representation significantly simplifies the process of generating a consistent 3D model for the input image using any pretrained diffusion model. Since the representation accounts for the reference view, the training simply has to focus on distilling the diffusion model prior for the other viewpoints. Therefore, we show that we can leverage a training algorithm that has minimal deviations from DreamFusion. We propose a few key improvements to the training algorithm which leads to more robust training and higher quality 3D reconstructions.

We demonstrate results using images gathered from multiple datasets including CO3D [16], ShapeNet [17], and images generated from pretrained generative models. ConRad produces 3D reconstructions that are consistently faithful to the input image while producing high quality full 3D reconstructions. In contrast to RealFusion and NeuralLift360, ConRad has fewer hyperparameters and does not rely on balancing the trade off between image reconstruction vs satisfying the diffusion prior, leading to a more robust training pipeline. We conduct a quantitative comparison on objects from the ShapeNet dataset and observe that our approach demonstrates significant gains in reconstruction quality compared to state-of-the-art methods.

## 2   Related Work

**3D Representations**   Over the last decade, research in Computer Vision and Graphics has led to development of efficient parametric representations of 3D scenes [13, 18, 19]. Traditional voxel

grid representations [20–22] maintain a dense grid of 3D scene parameters but consume a lot of memory. Neural Radiance Fields (NERFs) [13] offer a continuous parametric representation of the scene modeled as a neural network while requiring significantly lower memory while being computationally inefficient. Several approaches [23, 18] have been proposed to take advantage of both forms of representation. Instant-NGP [18] is one such approach that models a 3D scene using a multi-resolution hash-encoding followed by a shallow neural network. Our proposed ConRad representation builds on Instant-NGP but can be used with any radiance field.

**Learning to Estimate 3D Structure** To recover the full 3D structure from a single image, many works propose to learn category-specific 3D representations [24] then later fit to an input image. These approaches primarily relied on coarse representations like point clouds [24] and compositions of primitive shapes [25]. Furthermore, these methods focused on the geometry and could not infer the hidden appearance of the objects. NERFs demonstrated the capability to accurately infer novel viewpoints of a scene using densely sampled images. This has inspired numerous approaches to train similar radiance fields using sparse samples of a scene or object using semantic and geometric priors. [26–30] train models on large scale multi-view data to estimate radiance fields given a single or few images. The limited diversity of available 3D data restricts the generalizability of these approaches.

**Generating 3D from Pretrained 2D models** Large scale 2D image data is much more readily available compared to 3D or multi-view data. This has led to significantly larger advances in modeling semantics in images [4–6] and generative modeling of images [7, 8]. In light of this, several works have attempted to leverage the knowledge from these 2D models for the task of inferring the radiance fields from one or few images. Our proposed method belongs to this category of approaches since we rely on the distribution modeled by a pretrained image diffusion model.

DreamFields [31] generates 3D models from text prompts by optimizing the CLIP [4] distance between the NERF renderings and the textual CLIP embedding. DreamFusion [1] proposes an approach called Score Distillation Sampling (SDS) for distilling knowledge from a pretrained text-conditioned Diffusion Model [8] into a NERF representation (see Section 3 for more details). Magic3D [2] proposes a coarse-to-fine strategy that also leverages SDS to first train a NERF and then finetune a mesh representation. Similar to DreamFields, DietNERF [32] proposed to train a NERF on a few input images while enforcing consistency of CLIP [4] features in other unknown views. However, this approach fails to generate ε 3D objects using a single image input. Pretrained image generative models have also been used to infer radiance fields in existing work [33] but only focuses on estimating geometry of the regions visible in the input image.

RealFusion [14] and NeuralLift360 [15] are two recent methods that are very relevant to our proposed approach. Inspired by DreamFusion, these approaches leverage a pretrained Stable Diffusion model to infer the appearance and geometry of unknown viewpoints and optimize several additional objectives to enforce consistency to the input image in one reference viewpoint. In concurrent work, Nerdi[34] also follows a similar approach of using a reconsruction loss and depth loss for the known viewpoint, and uses diffusion model priors for unknown viewpoints. Unless properly tuned, these approaches lead to poor alignment between the input image and reconstructed 3D model (more details in Section 5). In contrast, ConRad focuses on modifying the underlying 3D representation to easily incorporate the input image and generates a more consistent 3D reconstruction.

## 3 Preliminaries

We first provide a concise summary of the prerequisite concepts from generative modeling of images and 3D objects that we build upon for ConRad.

**Diffusion Models** A diffusion model is a recently developed generative model that synthesizes images by iteratively denoising a sample from a Gaussian distribution. For training a diffusion model, noise is added to a real image $I$ iteratively for $T$ timesteps to synthesize training data $\{I_0, I_1, ..., I_T\}$. Specifically, the noised image for timestep $t$ can be computed as $I_t = \sqrt{\alpha_t}I + \sqrt{1 - \alpha_t}\epsilon$ where $\epsilon \in \mathcal{N}(\mathbf{0}, \mathbf{I})$ and $\alpha_t$ is predefined noising schedule. This data is used to a train a denoising model $\hat{\epsilon}_\phi(I_t, y, t)$ which estimates the noise $\epsilon$ using the noisy image $I_t$, the timestep $t$ and an optional embedding $y$ of the caption associated with the image. A pretrained diffusion model can be used to synthesize images by following the inverse process. First, a noise sample $I_T$ is drawn from $\mathcal{N}(0, \mathbf{I})$. Then $\hat{\epsilon}_\phi$ is iteratively used to estimate the sequences of images $\{I_{T-1}, I_{T-2}, ..., I_0\}$ where $I_0$ is the finally synthesized image.

**Neural Radiance Fields** A Neural Radiance Field (NERF) [13] is a parametric representation of a 3D scene as a continuous volume of emitted color $c$ and density $\sigma$. Formally, it can be written as a mapping $\mathcal{F}_\theta : (x) \rightarrow (c(x), \sigma(x))$ from the 3D coordinates of a point $x$, to it's associated color and density. These radiance fields allow rendering of 2D images by accumulating color over points sampled on camera rays. For a camera ray $\mathbf{r}(t) = \mathbf{o} + t\mathbf{d}$, the accumulated color is expressed as:

$$C(\mathbf{r}) = \int_t T(t)\sigma(r(t))c(r(t))dt \tag{1}$$

where $T(t) = exp(- \int_0^t \sigma(r(s))ds)$ is the accumulated transmittance of the volume along the ray. Learning an accurate representation $\mathcal{F}_\theta$ of a 3D scene generally requires supervision in the form of several ground truth samples of $C(\mathbf{r})$ *i.e.* several images of a scene taken from different viewpoints. Representing $\mathcal{F}_\theta$ as a neural network has been shown to demonstrate several desirable properties like the ability to synthesize novel views and realistic high-resolution renderings of complex scenes.

**Image Diffusion to Radiance Fields** DreamFusion proposed an approach to leverage a pretrained image diffusion model and a text prompt to optimize a 3D radiance field. The key idea is to optimize parameters $\theta$ such that the rendering of the radiance field from any viewpoint looks like a sample from the diffusion model. This is accomplished by randomly sampling camera poses $p$ during training, rendering images from the radiance field for these viewpoints $I_\theta^p$ and using a *Score Distillation Sampling* objective to optimize the radiance field. The gradient of the Score Distillation Sampling (SDS) objective $\mathcal{L}_{\text{SDS}}$ is defined as:

$$\nabla_\theta \mathcal{L}_{\text{SDS}}(\phi, I_\theta^p, y) = \mathbb{E}_{t,e}\Big[w(t)(\hat{\epsilon}_\phi(\sqrt{\alpha_t}I_\theta^p + \sqrt{1-\alpha_t}\epsilon, y, t) - \epsilon)\nabla_\theta I_\theta^p\Big] \tag{2}$$

where $w(t)$ is a timestep dependent weighting function. We refer the readers to [1] for the derivation of this objective. In practice, each update is computed using a randomly sampled timestep $t$ and noise sample $\epsilon$. Intuitively, this is equivalent to first perturbing the rendered image using $\epsilon, t$ and then updating the radiance field using the difference between the diffusion model estimated noise and $\epsilon$.

## 4 Method

### 4.1 Problem Setup

The goal of this work is to optimize a 3D radiance field $\mathcal{F}_\theta$ to capture the visible appearance and geometry of an object depicted in an input image, while inferring a realistic reconstruction of the unknown/hidden parts. Let the input image be represented by $\hat{I}$ and $\hat{p}$ be a reference camera pose (which can be arbitrarily chosen) associated with the image. Let $I_\theta^p$ be the image obtained as a differentiable rendering of $\mathcal{F}_\theta$ viewed from camera pose $p$. The optimal desired representation $F_{\hat{\theta}}$ should satisfy two criteria: (*i*) $I_{\hat{\theta}}^{\hat{p}} = \hat{I}$, and (*ii*) For all viewpoints $p$, $I_{\hat{\theta}}^p$ should be semantically and geometrically consistent.

A simple approach for satisfying requirement *(i)* could be to optimizing an $L_2$ distance objective $I_\theta^{\hat{p}}$ and $\hat{I}$. Furthermore, Score Distillation Sampling (see Section 3) can be used to satisfy requirement *(ii)* by optimizing randomly rendered viewpoints using diffusion model priors. This approach forms the basis of concurrent works RealFusion[14] and NeuralLift360[15]. We observe in experiments (Sec 5) that this approach often leads to misaligned final appearance of novel viewpoints.

### 4.2 ConRad: Image Constrained Radiance Fields

We propose a novel variant of neural radiance fields called Image Constrained Radiance Fields (ConRad) that allows us to effectively satisfy the two objectives. Intuitively, we wish to formulate a radiance field that accurately depicts the input image in one viewpoint while allowing us to optimize and infer the other viewpoints. First, we note that the input image depicts a "visible" part of an object's surface. Under some assumptions on the object properties[2], we can make inferences about the 3D space around this object. Any point that lies on/in front of the visible surface is known to us and does not need to be re-inferred in our radiance field. Specifically, points between the reference camera and the surface should have zero density, and points on the surface should have high density

---

[2] The proposed constraints rely on the assumption that the object is opaque.

and color equal to the corresponding pixel in the input image. Unfortunately, since we do not have access to the depth map corresponding to the reference view, it is not possible to infer which 3D points lie in front or behind the surface. As a workaround, we leverage the radiance field to obtain an estimate of the depth. Using this estimated depth, we propose to incorporate these observations into the color and density of the radiance field to leverage the input image as an explicit constraint.

Concretely, let the ray corresponding to pixel $(i, j)$ in the reference view be $\mathbf{r}_{\hat{p}}^{(i,j)}(t) = \mathbf{o}_{\hat{p}} + t\mathbf{d}_{\hat{p}}^{(i,j)}$. We define the *visibility depth* of this pixel $V_{\hat{p}}[i, j]$ as the value of $t$ such that

$$1 - \frac{\int_0^t T(t)\sigma(r_p^{(i,j)}(t))}{\int_{s=0}^{\infty} T(s)\sigma(r_p^{(i,j)}(s))} = \eta \tag{3}$$

where $\eta$ is a small value set to 0.1 in our experiments. Intuitively, the visibility depth for each pixel is a point on the ray beyond which the contribution of color is minimal (less than 10%).

Let $Q_p : \mathbb{R}^3 \to [-1, 1] \times [-1, 1]$ be the camera projection matrix corresponding to projection from world coordinates to the normalized reference image coordinates. We can reformulate the color $c(x)$ of the radiance field $\mathcal{F}_\theta$ as follows:

$$v_x = \mathbb{1}(\; ||x - \mathbf{o}_{\hat{p}}|| < V_{\hat{p}}[Q_{\hat{p}}(x)]\;) \tag{4}$$

$$c'(x) = v_x * I^{\hat{p}}(Q_{\hat{p}}(x)) + (1 - v_x) * c(x) \tag{5}$$

where we use bilinear interpolation to compute pixel values $I^{\hat{p}}(Q_{\hat{p}}(x))$. This constraint enforces the appearance of the reference viewpoint to explicitly match the image. Since we only estimate the depth of each pixel based on a potentially incorrect density field, we enforce that all points in between the camera and the estimated surface along the ray should have color equal to the corresponding pixel. As shown in Figure 1, this still constrains the reference view to match the input image but allows the density to be optimized through our training process.

Additionally, the foreground mask of the input image informs the density of the radiance field. We estimate the binary foreground mask $\hat{M}$ using a pretrained foreground estimation method[35]. We know that any point on the reference rays corresponding to the background pixels should have zero density. We reformulate the density $\sigma(x)$ as:

$$m_x = \hat{M}[Q_{\hat{p}}(x)] \tag{6}$$

$$\sigma'(x) = m_x * \sigma(x) \tag{7}$$

In summary, ConRad is the radiance field defined by the constrained color $c'(x)$ and density $\sigma'(x)$.

### 4.3 Optimizing Image Constrained Radiance Fields

In this section, we present our approach for training a ConRad representation using a single input image $\hat{I}$. First, we preprocess the input image to extract several supervisory signals. Since we wish to leverage a text-conditioned diffusion model, we need to generate text embeddings for a caption of the input image. We use the caption "a photo of a <token>" where the embedding of the special token is inferred using Textual Inversion[36]. In order to learn an embedding that accurately captures the input image, we pass in multiple synthetically generated augmentations of the input image by randomly cropping, resizing and blurring it. Due to space constraints, we presented more details in the supplementary material. We also estimate a depth map for the reference view $\hat{D}$ using MiDaS[37].

**Diffusion Prior** One of the key advantages of ConRad is the simplicity of the training pipeline. Since ConRad already incorporates the appearance of the reference view, the training algorithm has minimal deviations from DreamFusion. We simply compute SDS updates on random viewpoints using the pretrained Diffusion Model conditioned on the previously obtained text embeddings and update the radiance field to infer the unknown density and color regions.

**Depth Loss** Since the depth of the reference viewpoint is unknown, we find that providing additional supervision through the estimated reference depth map $\hat{D}$ sometimes improves the results of our training. The preprocessed depth estimate $\hat{D}$ only provides relative depth values[38]. Therefore, the ground truth depth should be close to $\hat{D}$ up to a scalar scaling and translation factor. We accommodate

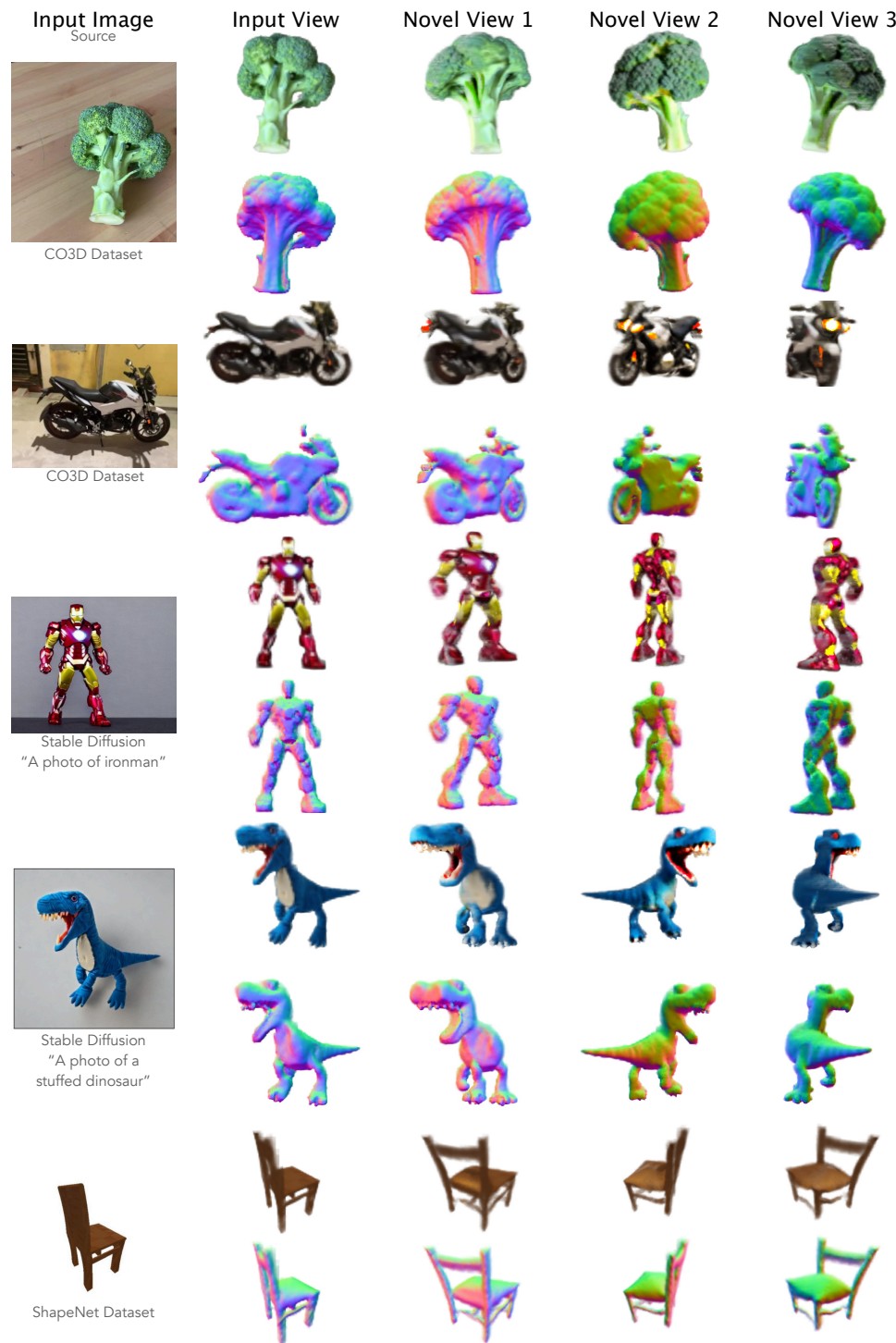

**Figure 2: ConRad Image-to-3D Reconstruction:** Here we visualize the 3D structures generated by our proposed approach using images taken from different sources. We observe that the proposed ConRad representation leads to high quality realistic reconstructions of the objects while completely preserving input image details. Please see Appendix for comparison to RealFusion and NeuralLift360 on these input images.

this in the depth loss $\mathcal{L}_{\texttt{dep}}$ by formulating it as the Pearson Correlation coefficient between $D$ and $\hat{D}$. NeuralLift360[15] adopts a similar idea and uses a ranking loss for providing depth supervision.

**Warm Start** The visibility and mask values in Eq 4 & 6 are non-differentiable and also cause a sharp boundary in the radiance field. For some input objects, we observe that this disrupts training. This can easily be resolved by performing a "*warm start*". We multiply the visibility and mask scores in Eq 4 & 6 with a scalar $\alpha$ that is linearly annealed from 0 to 1 over the first 50% of the updates and then kept constant at 1.

## 5 Experiments and Results

### 5.1 Implementation Details

The simplicity of Image Constrained Radiance Fields representations facilitates a straightforward implementation of the training process. We use Instant-NGP[18] as the base representation comprising of a shared multi-resolution hash encoding and two separate small MLPs for the unconstrained color $c(x)$ and density $\sigma(x)$ respectively. For all experiments in this paper, we choose the reference view camera to lie at a distance of 3.2 from origin, azimuth angle 0 and elevation 0 except where specified. We precompute and store rays for the camera corresponding to this reference view. Before each update, we estimate the Visibility Depth (Eq 4) by evaluating on points along these rays and choosing the closest solution. For each update, a random camera pose $p$ is obtained by uniformly sampling elevation angle in $[-15°, 45°]$, azimuth angle in $[0°, 360°]$ and distance to origin in $[3, 3.5]$. We then render the image $I_p$ for this viewpoint using the constrained color $c'(x)$ and density $\sigma'(x)$. We also compute the estimated reference view depth $D_{\hat{p}}$ using the precomputed rays and radiance field density $\sigma'(x)$. We compute the SDS sampling update (Eq 2) using $I_p$ and the gradient of the Depth Loss $D_{\hat{p}}$ to update all the parameters. Additionally, the regularizations proposed in [1] are adopted to enforce smoothness of surface normals and encourage outward facing surface normals in the radiance field. We keep all the hyperparamters unchanged for all experiments except when explicitly indicated (Sec 5.4). Please refer to the supplementary material for details of regularizations and their associated weights, hyperparameters of Instant-NGP, optimizer hyperparameters and pseudo-code of the training algorithm.

Note that we do not need to render the image of the reference view during training. This eliminates the need for tuning viewpoint sampling strategies and additional reference loss hyperparameters, leading to a simpler and more robust training pipeline. Computation of visibility depth also does not significantly increase GPU memory consumption since we do not compute its gradients.

### 5.2 Visualizing 3D Reconstructions

We first visualize the final 3D reconstructions of ConRad representations trained on single images taken from different sources. In Figure 2, we present the RGB and surface normal reconstructions of the reference view and three novel viewpoints. Observe that the final representation always accurately reconstructs the input viewpoint due to the constraints incorporated in ConRad. Furthermore, the novel viewpoints generally depict a realistic reconstruction of the input object.

We also observe that the reconstructions are faithful to the specific instance of the object, presenting a smooth transition from reference viewpoint to unobserved viewpoints. This can be attributed to two aspects of the model: (i) The textual inversion embedding passed to the diffusion model captures the details of the object and (ii) Since ConRad depicts the input image accurately from beginning of training, it provides a strong prior for the appearance for the diffusion model, especially around the reference viewpoint. In Figure 2, "Novel View 1" presents a viewpoint close to the reference view.

Our proposed approach can also be used to generate 3D model from text prompts. To accomplish this, we can leverage pretrained Stable Diffusion to first generate a 2D image from the text prompt and then use ConRad to learn a 3D reconstruction of this object. Figure 2 Rows 3 & 4 demonstrate this capability on two prompts.

### 5.3 Comparison to existing work

**RealFusion**[14] relies on alternating between rendering the reference viewpoint and a random viewpoint. For the reference viewpoint, additional mean squared error objectives are used to distill

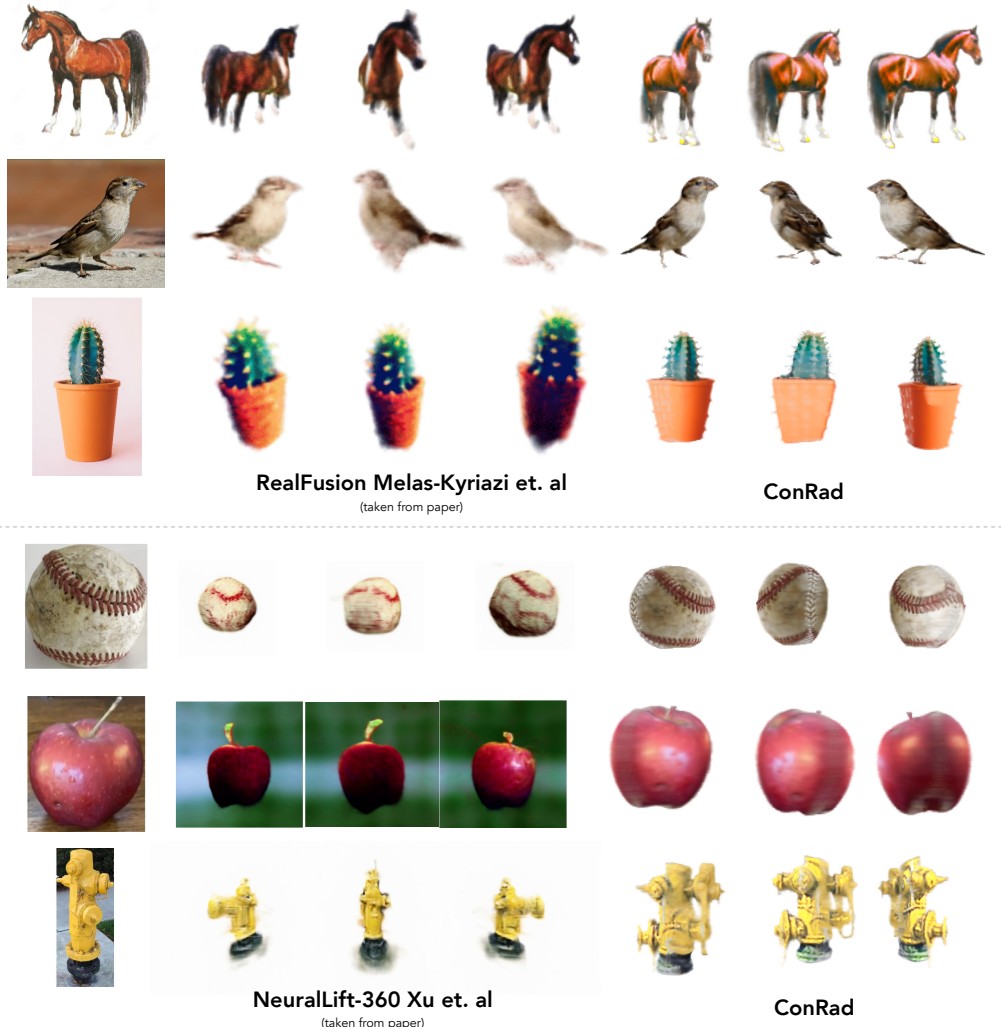

**Figure 3: Qualitative comparisons:** We perform qualitative comparisons of our approach to state-of-the-art image to 3D generation methods. For fair comparison and to demonstrate robustness, we evaluate on images taken from the respective papers. We observe that ConRad is able to generate higher quality 3D models while more accurately preserving input image details.

the reference RGB and foreground mask values into the 3D representation. For other viewpoints, Score Distillation Sampling is used to update the representation. Similar to our approach, textual inversion is used to compute the conditioning text embedding for Stable Diffusion.

**NeuralLift360**[15] proposes a similar approach to RealFusion. In addition to reference RGB and foreground mask, NeuralLift360 also utilizes an estimated depth map and the CLIP features of the input image as supervision. A ranking loss is used to account for the scale and translation ambiguity in the estimated depth map. This is similar in spirit to our proposed depth objective in Section 4.3. NeuralLift360 also encourages the CLIP features of renderings from all viewpoints to be consistent with the input viewpoint.

**Qualitative Comparisons** In Figure 3, we present qualitative comparisons to ConRad. While both RealFusion and NeuralLift360 released official implementations, we observed that they require tuning of hyperparameters or fail to generate meaningful reconstructions for several objects. For fair comparison, we evaluate our method on images taken from the respective papers. We observe

ConRad can better capture the details of the input image consistently. In contrast, RealFusion and NeuralLift360 often generate a similar instance of the same category with a different appearance.

**Evaluation Metrics** The task of image to 3D generation is difficult to quantitatively evaluate due to the inherent ambiguity in the expected output *i.e.* there are several possible realistic reconstructions of the same object. NeuralLift360[15] evaluates the ability to capture semantic content by measuring the CLIP feature[4] distance between all viewpoints of the generated object and the single input reference image. We build upon this idea and propose metrics to evaluate the ability to generate different viewpoints of an object.

**Table 1: Quantitative Comparisons:** We perform quantitative comparisons using 3D object data of 20 objects depicting instances of 10 categories taken from the ShapeNet dataset. We evaluate the CLIP semantic similarity of rendered object viewpoints to ground truth viewpoint samples. We demonstrate that generating 3D models with ConRad leads to significant improvements over state-of-the-art across all metrics.

| Method | All Views | | | Near Reference | | |
|---|---|---|---|---|---|---|
| | $d_{ref}$ | $d_{all}$ | $d_{oracle}$ | $d_{ref}$ | $d_{all}$ | $d_{oracle}$ |
| PointE[39] | 0.496 | 0.482 | 0.453 | 0.496 | 0.499 | 0.399 |
| RealFusion[14] | 0.495 | 0.486 | 0.460 | 0.483 | 0.482 | 0.464 |
| NeuralLift-360[15] | 0.528 | 0.516 | 0.498 | 0.543 | 0.544 | 0.534 |
| ConRad (ours) | **0.332** | **0.316** | **0.273** | **0.286** | **0.257** | **0.230** |

First, we render 20 objects from 10 categories of the ShapeNet[17] dataset viewed from 68 different camera poses to create a ground truth (GT) set. We choose a front-facing view from each object as input to an Image-to-3D approach. We then render the generated object from the same 68 camera poses. Due to ambiguity of depth, corresponding camera poses between GT and rendered images could depict very different object poses. Using these two sets of images, we compute three metrics – $d_{ref}$ is the mean CLIP feature distance between the reference image and all the rendered viewpoints (same as [15]). $d_{all}$ is the mean CLIP feature distance between all pairs of GT and rendered images. $d_{oracle}$ is the solution to a linear sum assignment problem[40] where the cost of assigning a GT view to a rendered image is the CLIP feature distance between them. This evaluates the ability of the representation to generate images as diverse as the ground truth while preserving semantic content.

We evaluate these three metrics for the two sets of 68 images ("All Views"). We also evaluate on a subset of camera poses that lie within a $15°$ elevation change and $45°$ azimuth change ("Near Reference") giving us 15 images each for ground truth and rendered images. This measures the semantic consistency in viewpoints where parts of the input image are visible.

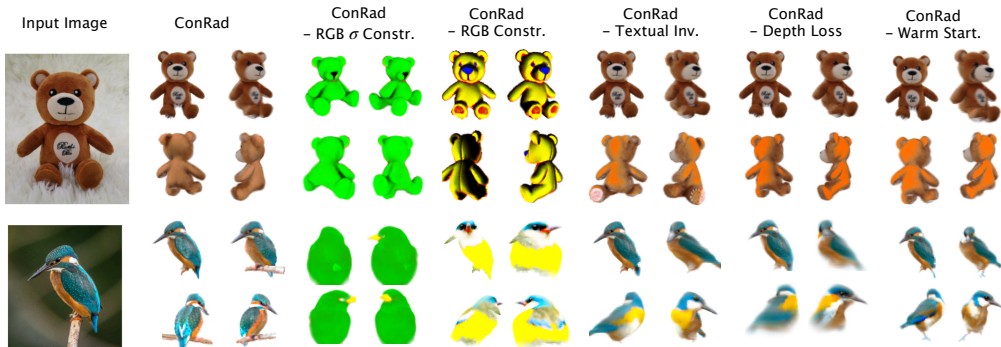

**Figure 4: Ablation Study:** We present a qualitative investigation of the importance of various components proposed in our approach on two prototypical examples. The components are arranged left to right in decreasing order of importance for final reconstruction quality.

## 5.4 Analysis of ConRad Components

**Quantitative Evaluation**   In Table 1, we compare ConRad to RealFusion, NeuralLift360, and PointE. PointE[39] is a point cloud diffusion model was trained on several million 3D models. It can directly generate point clouds using CLIP features of the input image. Since there is no corresponding reference view in the output, we synthesize 150 random views for PointE instead of the 68 chosen views. For all methods, we use the implementations and hyperparameters provided by the authors. We observe that ConRad outperforms these methods across all the metrics by a significant margin. We present a per-object breakdown of these metrics in the supplementary material.

We now investigate the various components of ConRad to understand their significance. In Figure 4, we present a visualization of the effect of removing each component on two typical examples. In most experiments, we observe that performing a *warm start* is not necessary but leads to crisper final 3D structure. We also observe that removing the Depth Loss significantly reduces computational cost, but this leads to incorrect 3D structure for some objects (see row 2 of Figure 4). Textual Inversion is crucial to maintain consistent appearance of the object from all viewpoints. Moreover, removing the color constraint (Eq 4) generally still leads to a realistic 3D model but depicts an arbitrary object. Finally, removing both color and density constraints (still using the depth loss) leads to very unrealistic objects.

## 5.5 Failure Cases and Limitations

While ConRad significantly improves the simplicity and robustness of learning the 3D structure, it occasionally demonstrates issues that are common to Image-to-3D methods. In Figure 5, we present two typical examples of failure cases. Row 1 demonstrates the Janus effect, where the final 3D model has two faces. We also observe that performing score distillation sampling using Stable Diffusion leads to saturated colors in the 3D model. Leveraging textual inversion embeddings mitigates this issue to some extent, but still occasionally leads to incorrect appearance in novel viewpoints, as demonstrated in Row 2. We also observe that for a few objects, parts of the generated 3D object is semi-transparent (see videos in supplementary material).

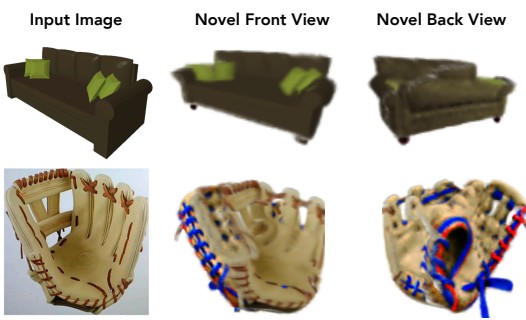

**Input Image**   **Novel Front View**   **Novel Back View**

**Figure 5: Failure Cases:** On some objects, our proposed approach suffers from the Janus effect and is affected by the bias of Stable Diffusion to produce saturated colors.

## 6   Conclusion

In this work, we present a novel radiance field variant, ConRad, that significantly simplifies the training pipeline for generating 3D models from a single image. By explicitly incorporating the input image into the color and density fields, we show that we can eliminate tedious tuning of additional hyperparameters. We demonstrate that ConRad representations lead to more accurate depiction of the input image, produce higher quality and more realistic 3D models compared to existing work.

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

# A   Additional Implementation Details

Here we present more details about the implementation of ConRad representation to facilitate reproducibility of results.

**Base Representation**   As indicated in the main text, we use the Instant-NGP[18] representation followed by two MLPs to model the density $\sigma(x)$ and color $c(x)$. For the multi-resolution hash encoding of Instant-NGP representation, we use 16 levels with a 2-dimensional encoding at each level. The Instant-NGP encoding is passed to two 3-layer MLPs with a hidden dimension of 64. The color MLP output is passed through a sigmoid activation to obtain the RGB values additionally. The density MLP output is passed to the exponential $e^x$ activation to obtain the density value.

**Regularization Losses**   In addition to SDS and the Depth Loss, we use three regularizations proposed in [1] to produce coherent objects. We encourage the radiance field to have either very low or very high density at any point along the rays by using the following entropy regularizer:

$$\mathcal{L}_{ent} = \sum_x -\alpha(x) * \log \alpha(x) - (1 - \alpha(x)) \log(1 - \alpha(x)) \tag{8}$$

where $\alpha(x)$ is the rendering weight at point $x$.

We encourage the surface normals in each rendered view to point towards the camera using the orientation regularizer:

$$\mathcal{L}_{orient} = \sum_x \alpha(x) * \max(< \mathbf{n}(x), \mathbf{d} >, 0)^2 \tag{9}$$

where $\mathbf{n}$ is the normal at the point $x$ computed using the finite differences method with the density $\sigma(x)$ and $\mathbf{d}$ is the viewing direction.

Finally, we encourage the smoothness of normals computed at each point along the rays using the smoothness regularizer:

$$\mathcal{L}_{smooth} = \sum_x |\mathbf{n}(x) - \mathbf{n}(x + \delta)| \tag{10}$$

where $\delta$ is a random perturbation for each point with a maximum perturbation of 0.01 along each axis.

---

**Algorithm 1:** Optimizing a ConRad representation to reconstruct 3D from a single image.

---

**Data:** Image $\hat{I}$, Estimated Depth $\hat{D}$, Foreground Mask $\hat{M}$, Reference Pose $\hat{p}$

1  Initialize ConRad $\mathcal{F}_\theta$;
2  Compute reference rays $r_{\hat{p}}$;
3  Compute text embeddings $t$ using textual inversion on image $\hat{I}$;
4  **for** $i = 0$ **to** 5000 **do**
5  $\quad$ $\hat{V} \leftarrow$ Visibility Depth using $r_{\hat{p}}$ and $\sigma_\theta(x)$ (Equation 3);
6  $\quad$ $D_{\hat{p}} \leftarrow$ Radiance Field Depth along rays $r_{\hat{p}}$ using $\sigma'(x)$ ;
7  $\quad$ $\mathcal{L}_{dep} \leftarrow$ Pearson Correlation between $D_{\hat{p}}$ and $\hat{D}$;

8  $\quad$ $p \leftarrow$ sample random camera pose;
9  $\quad$ $r_p \leftarrow$ compute rays for viewpoint $p$;
10 $\quad$ $I_p \leftarrow$ Render reference viewpoint image using $r_{\hat{p}}, c'(x), \sigma'(x)$;

11 $\quad$ $\nabla_\theta \mathcal{L}_{\text{SDS}}(\phi, I_\theta^p, t) \leftarrow$ Using Stable Diffusion, text embedding $t$ and $I_p$ (Eq. 2);
12 $\quad$ $\mathcal{L}_{orient} \leftarrow$ Compute orientation regularization along $r_p$;
13 $\quad$ $\mathcal{L}_{smooth} \leftarrow$ Compute smoothness regularization along $r_p$;
14 $\quad$ $\mathcal{L}_{ent} \leftarrow$ Compute entropy regularization along $r_p$;
15 $\quad$ $\theta \leftarrow \theta - \eta(\nabla_\theta \mathcal{L}_{\text{SDS}}(\phi, I_\theta^p, t) + \nabla_\theta(10 * \mathcal{L}_{depth} + 0.01 * \mathcal{L}_{ent} + 0.01 * \mathcal{L}_{orient} + 10 * \mathcal{L}_{smooth}))$;

---

**Loss Weighting**   The above losses are all jointly optimized using the following weighted sum:

$$\mathcal{L} = \mathcal{L}_{\text{SDS}} + 10 * \mathcal{L}_{depth} + 0.01 * \mathcal{L}_{ent} + 0.01 * \mathcal{L}_{orient} + 10 * \mathcal{L}_{smooth} \tag{11}$$

In experiments, we observed that the orientation and entropy regularizer had minimal effect on the final output and could be turned off on most input images without any loss in quality. However, we retain these regularizers in all experiments to ensure similarity to DreamFusion.

**Textual Inversion**   Since we aim to use only an image as input, we need to synthesize a text prompt to pass as conditioning to the text-conditioned diffusion model. Additionally, we need the text-prompt to be sufficiently descriptive to capture all the details of the specific object instance. Most pretrained captioning algorithms provide a coarse caption that doesn't capture such details. Therefore, in recent research, Textual Inversion[36] proposed an approach that uses input images and a diffusion model to infer the text embedding of a special token ("<token>"). This token along with its learned embedding can be passed to the diffusion model to synthesize novel images of the same object. This approach generally requires 3 to 5 images to accurately capture an object. Since we have access to only one image, we rely on synthetically augmenting the image to run Textual Inversion. Specifically, we generate images by randomly flipping horizontally with probability 0.5, extract a random crop covering 50% to 100% of the image, Gaussian blurring the image with a kernel size 5 and standard deviation randomly sampled in $[0.1, 2]$ and jittering the hue, saturation, contrast and brightness by a random value in $[0, 0.1]$. Furthermore, we perform classification of the input image using a pretrained CLIP model[4]. The text embedding of the obtained class label is used to initialize the embedding of the special token "<token>".

**Optimization**   We use the Adan optimizer with 0.005 learning rate and weight decay of 0.00002. We use a batch size of 1 for each update *i.e.* one random viewpoint. We keep the learning rate fixed and train the representation for 5000 updates. The optimization process takes approximately 20 minutes on a single A100 GPU.

## B   Additional Quantitative Evaluation

In Sec 5.3 of the main text, we present quantitative comparisons of our work to RealFusion[14], NeuralLift-360[15] and Point-E[39] on objects from the ShapeNet[17] dataset. Here we present more details on this evaluation to facilitate easy reproduction.

**Evaluation Data**   We choose 20 objects from 10 categories. While we randomly sampled objects, we manually removed some non-prototypical objects like a triangular bed. In Figure 6, we present the chosen input image for each object, the ShapeNet object ID and category.

**Table 2: Quantitative Comparisons:** We present quantitative comparisons using 3D object data of 20 objects the ShapeNet dataset. We present an object-wise breakdown of the performance for each method. The performance is evaluated on all (68) views of the object. Please refer to the main text for more details on the evaluation metrics.

| Category | Object ID | $d_{ref}$ | | | | $d_{all}$ | | | | $d_{oracle}$ | | | |
|---|---|---|---|---|---|---|---|---|---|---|---|---|---|
| | | RealFusion | NeuralLift-360 | Point-E | ConRad | RealFusion | NeuralLift-360 | Point-E | ConRad | RealFusion | NeuralLift-360 | Point-E | ConRad |
| airplane | 2c9797... | 0.418 | 0.753 | 0.412 | **0.251** | 0.462 | 0.833 | 0.441 | **0.263** | 0.432 | 0.833 | 0.420 | **0.236** |
| airplane | 7d89d6... | 0.539 | 0.543 | 0.426 | **0.246** | 0.658 | 0.676 | 0.513 | **0.279** | 0.639 | 0.668 | 0.487 | **0.238** |
| bench | 5d9880... | 0.642 | 0.737 | 0.507 | **0.374** | 0.583 | 0.710 | 0.457 | **0.355** | 0.557 | 0.692 | 0.427 | **0.297** |
| bench | 62cc45... | 0.575 | 0.585 | 0.395 | **0.272** | 0.599 | 0.598 | 0.408 | **0.309** | 0.578 | 0.592 | 0.366 | **0.249** |
| bus | 2ba84d... | 0.533 | 0.541 | 0.564 | **0.335** | 0.492 | 0.548 | 0.501 | **0.329** | 0.461 | 0.537 | 0.469 | **0.259** |
| bus | 642b3d... | 0.485 | 0.531 | 0.571 | **0.356** | 0.500 | 0.474 | 0.574 | **0.340** | 0.462 | 0.416 | 0.540 | **0.275** |
| bus | 6cc0a9... | 0.654 | 0.588 | 0.582 | **0.340** | 0.610 | 0.534 | 0.523 | **0.295** | 0.587 | 0.502 | 0.490 | **0.252** |
| car | 1abeca... | 0.490 | **0.458** | 0.578 | 0.469 | 0.487 | 0.455 | 0.551 | **0.401** | 0.460 | 0.411 | 0.517 | **0.350** |
| car | 35155f... | 0.514 | 0.396 | 0.527 | **0.368** | 0.504 | 0.340 | 0.527 | **0.311** | 0.481 | 0.300 | 0.501 | **0.265** |
| car | 9807c1... | 0.543 | 0.449 | 0.582 | **0.371** | 0.503 | 0.377 | 0.561 | **0.281** | 0.482 | 0.347 | 0.535 | **0.238** |
| chair | 8ce2e4... | 0.562 | 0.837 | 0.508 | **0.370** | 0.470 | 0.773 | 0.432 | **0.304** | 0.446 | 0.773 | 0.395 | **0.269** |
| chair | ea9181... | 0.548 | 0.517 | 0.485 | **0.259** | 0.522 | 0.495 | 0.471 | **0.248** | 0.500 | 0.490 | 0.452 | **0.208** |
| flowerpot | 8e7642... | 0.338 | 0.505 | 0.497 | **0.319** | **0.299** | 0.483 | 0.481 | 0.307 | **0.283** | 0.469 | 0.467 | 0.290 |
| flowerpot | f25999... | 0.399 | 0.252 | 0.467 | **0.198** | 0.446 | 0.288 | 0.491 | **0.227** | 0.425 | 0.264 | 0.471 | **0.186** |
| guitar | 4c082a... | 0.440 | 0.402 | 0.558 | **0.335** | 0.363 | 0.340 | 0.474 | **0.304** | 0.336 | 0.313 | 0.435 | **0.274** |
| motorbike | 61b17f... | 0.407 | 0.586 | 0.482 | **0.327** | 0.391 | 0.597 | 0.508 | **0.297** | 0.353 | 0.593 | 0.483 | **0.252** |
| motorbike | 6e3761... | 0.409 | 0.395 | 0.521 | **0.270** | 0.410 | 0.348 | 0.512 | **0.267** | 0.379 | 0.314 | 0.491 | **0.230** |
| sofa | 6f8494... | 0.551 | 0.556 | 0.458 | **0.328** | 0.517 | 0.516 | 0.424 | **0.337** | 0.493 | 0.512 | 0.386 | **0.281** |
| table | 5d9f9e... | 0.481 | 0.471 | 0.387 | **0.294** | 0.503 | 0.478 | 0.395 | **0.297** | 0.477 | 0.474 | 0.365 | **0.252** |
| table | bdd12e... | **0.380** | 0.458 | 0.406 | 0.560 | 0.399 | 0.457 | **0.394** | 0.568 | **0.376** | 0.451 | **0.370** | 0.562 |

**Detailed Quantitative Results**   In Table 2, we present an object-wise breakup of the results presented in the main text. We observe that for *most* objects, ConRad based training leads to significantly improved results on all metrics.

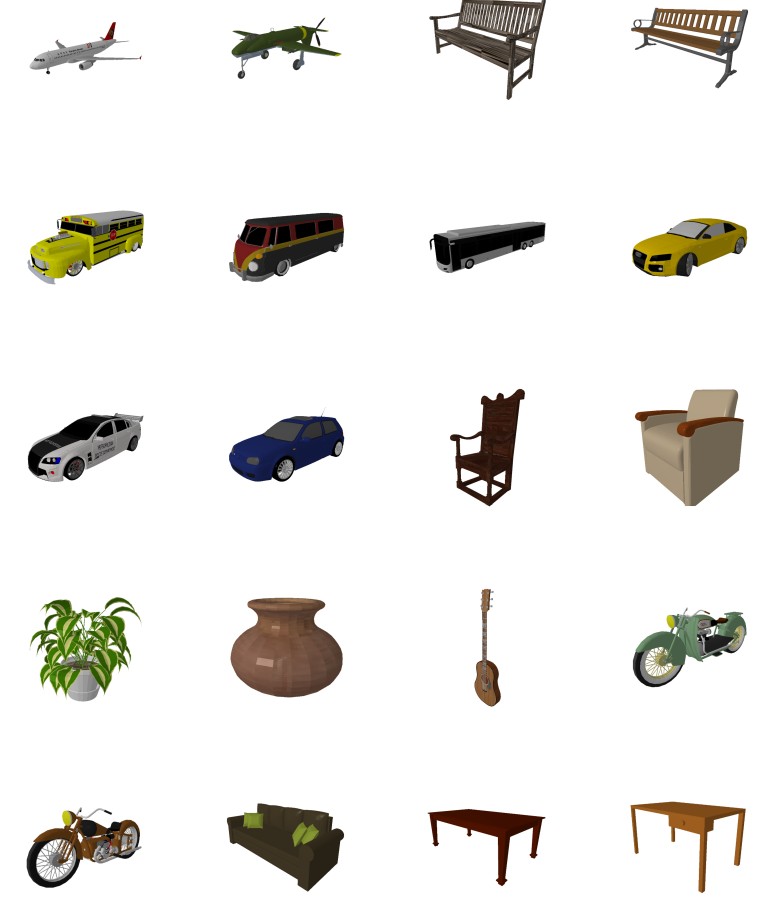

**Figure 6: ShapeNet Evaluation Data:** We present the 20 input images used to evaluate performance ConRad and existing works in this domain.

# C Additional Qualitative Comparisons

In Figure 2, we present visualizations of 3D models produced by ConRad. Here we present visualizations for the 3D models produced by RealFusion[14] and NeuralLift360[15] using the same input images in Figure 7. We observe that both methods generally produce low quality generations when arbitrary images are used as inputs. We found that NeuralLift360 was successful more frequently compared to RealFusion but produced inferior 3D models compared to ConRad.

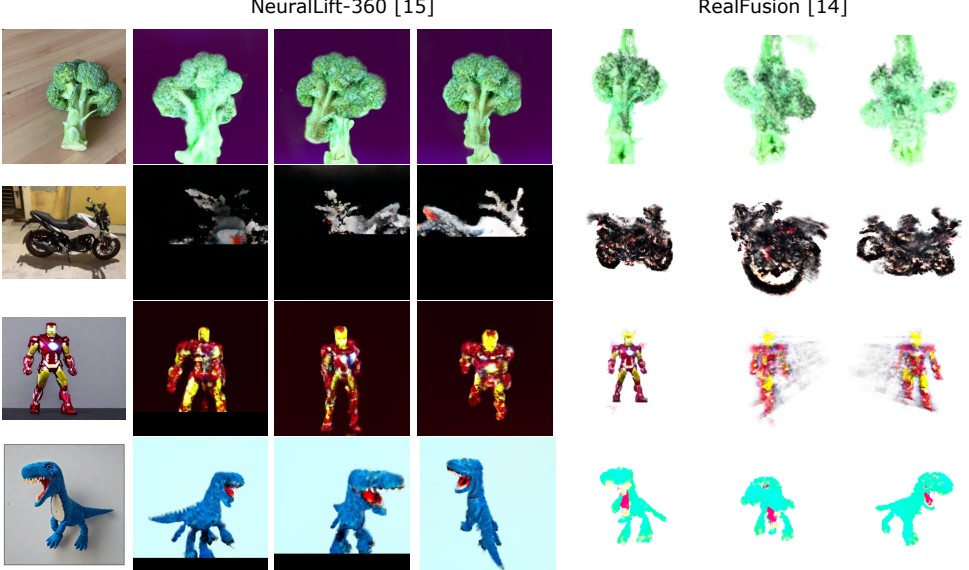

**Figure 7: Additional Qualitative Baseline Results:** We present additional qualitative baseline results using reference images presented in the main paper. For both NeuralLift-360 and RealFusion, we use the official implementations released by the authors.

