# OpenReview forum: "ConRad: Image Constrained Radiance Fields for 3D Generation from a Single Image"
_NeurIPS.cc/2023/Conference — NeurIPS 2023 poster_

### Official Review · Reviewer_mCGc · 2023-07-03

**Soundness:** 3 good
**Presentation:** 2 fair
**Contribution:** 2 fair
**Rating:** 5
**Confidence:** 4

**Summary:**

This paper proposes image constraint neural radiance field, a representation that takes a reference image into account. Such a representation helps the task of 3D reconstruction from single image with the guidance from pretrained diffusion models. Experiments demonstrates the proposed method can help boost the quality of the reconstructed objects.

**Strengths:**

1. The proposed method is simple and effective.
2. The experiments show the effectiveness of the proposed image constraint neural radiance field.

**Weaknesses:**

1. For viewpoints other than the reference view, the shapes and images are blurry. The proposed method can only enforce the appearance under the reference view to be realistic, giving limited improvement to views that do not have overlap with the reference view.
2. In line 49, '…can explicitly capture an input image without any training' which is ambiguous. The underlying depth of the input image still requires depth loss to constrain the learning of the radiance field.
3. The proposed methods claims to fully utilize the information from the reference view compared to the previous methods(L55-57). The quality is better than the baselines, but how about the efficiency? Does it take shorter time to reach a good reconstruction result?
4. There are some typos in the paper, e.g., L54 'a' should be removed; Eq.(4) 'o_p' should be bold to align with L172

**Questions:**

Please refer to weaknesses.

**Limitations:**

The authors have mentioned some of the limitations in Sec 5.5.

---

> ### Author Rebuttal · Authors · 2023-08-10
>
> We thank Reviewer mCGc for the positive review and constructive feedback. We clarify the concerns raised by the reviewer here.
>
> > 1. For viewpoints other than the reference view, the shapes and images are blurry.
>
> Compared to the reference view, the renderings do appear less crisp in novel viewpoints. However, this is a limitation of most image to 3D approaches. In fact, we demonstrate in results that compared to existing works, our approach produces higher quality reconstructions even in novel viewpoints (Figure 3 and Table 1).
>
> > 2. In line 49, '…can explicitly capture an input image without any training' which is ambiguous. The underlying depth of the input image still requires depth loss to constrain the learning of the radiance field.
>
> We will rephrase this in the paper to make it clearer. This statement is referring to the rendering of the radiance field from the reference viewpoint. The proposed method can indeed capture an input image *in the reference viewpoint* without any training. The constraints applied on the radiance field are instantaneous and do not require any optimization. For example, in Figure 1, we show a visualization of the constraints applied using the image of a chair. As we show on the right, the rendering from the reference viewpoint is accurately capture without any training.
>
> > 3. The quality is better than the baselines, but how about the efficiency? Does it take shorter time to reach a good reconstruction result?
>
> Intuitively, we hoped to see improvements in efficiency since ConRad captures the reference view accurately. However, in practice we did not observe convincing speed improvements in the number of updates required. Therefore, we do not make this claim in the paper. We keep the number of updates equal to RealFusion which works well for all objects. Both methods take approximately 20 minutes on a single A100 GPU. NeuralLift-360 uses twice the number of updates and takes 1 hour on the same hardware setup. We will add these details to the paper.
>
> > 4. There are some typos in the paper, e.g., L54 'a' should be removed; Eq.(4) 'o_p' should be bold to align with L172
>
> Thank you for pointing this out. We will make these changes.

---

> > ### Comment · Reviewer_mCGc · 2023-08-19
> >
> > Thanks for the explanations. The authors have addressed my concerns and I will update my ratings.

---

### Official Review · Reviewer_1JJW · 2023-07-04

**Soundness:** 4 excellent
**Presentation:** 4 excellent
**Contribution:** 4 excellent
**Rating:** 7
**Confidence:** 4

**Summary:**

This paper proposes a novel parametrization for NeRF designed to facilitate the task of single image (+ foreground mask) to 3D model generation. The authors modify the volumetric rendering equation of the NeRF volume to include explicit constraints given by the single available view. In particular, by construction, points intersected by rays corresponding to the background of the conditioning image will have their density set to zero and points intersected by rays corresponding to the foreground will have an RGB color equal to the one in the conditioning image (instead of the one encoded in the radiance field). Given this modified NeRF rendering procedure the authors use textual inversion + an SDS loss similar to Dreamfusion to optimize a full 3D model. The proposed method can be used both to generate plausible 3D models of real objects from a single view as well as to generate 3D models from pure text using a text2image model to get the conditioning view. The method is compared to concurrent works and achieves better quality and more crisp 3D models.

**Strengths:**

+ The proposed solution is both simple and elegant while achieving convincing results. The modification to the rendering equation guarantees, by construction, that the single input view is going to be respected. This allows the capacity of the NeRF volume to focus on modeling the missing parts and recovering the geometry of the first view (when not using the monocular depth loss).

+ Great presentation, the paper is well written and quite self contained. Most details regarding the implementation of the method are provided as part of the appendix.

+ I appreciated the effort of the authors in Sec. 5.3 Tab. 1 to try to propose a novel metric for evaluating the quality of the inferred 3D representations in terms of their 3D consistency.


**Weaknesses:**

## Major

I have not found major weaknesses in the work.

## Minor

a. **Scale Ambiguity**: The scale of the 3D reconstructed model is inherently ambiguous due to the unknown depth associated with the first view. This means that depending on how the density of the first view converges the same 3D object could be represented as a big object far away or a small object close to the original camera. Of course this, in turns, implies that moving the camera once the model is fitted will have drastically different effects. This is also recognized by the authors between line 287 and 290. This is an inherent ambiguity of any methods based on a single view but I wonder if a simple regularization to push the density of the NeRF towards the central area of the volume could have helped to standardize the scale of the fitted 3D models.

b. **Possibly entangled text inversion**: to keep faithful details of the object the authors propose to perform textual inversion of a Stable Diffusion model to find a text token corresponding to the appearance of the object they are trying to reconstruct. To do so they apply various augmentation to the single view of the object to generate a training set for textual inversion and directly optimize an input token. Since all the images used for textual inversion are generated from a single seed image the textual inversion token might pick up on specific details of the single view available rather than on the specific object that the authors are trying to reconstruct. An example of unwanted entangle representation could be to pick up on  the background of the single view. Randomly cutting and pasting the foreground object into random positions over random backgrounds might be a viable solution to reduce this risk.

c. **Some ad hoc components per experiments**: From the manuscript it seems that the authors used the depth loss and the warm start only for some experiments but not for all of them. This makes evaluating the experimental evaluation a bit more confusing. If the two additional components do not have any negative impact on the final performance also for models which would not need them (as it seems the case from the ablations in Fig. 4) I would suggest just presenting all results with both components switched on for all experiments. If instead in some cases those have a negative effect I would mention it explicitly in the manuscript.


**Questions:**

## Questions

1. How are you handling the background when training with the SDS? The NeRF being optimized does not have any background, are you adding a random one before feeding the image to Stable Diffusion to compute the SDS?

2. Can you clarify any possible misunderstandings with respect to what I wrote in weakness “b”?

3. The procedure used to initialize the text token described between line 38 and 40 of the appendix it’s not very clear. How do you perform classification using CLIP in this context?

## Typos

* line 54: “the process of a generating” → “the process of generating”


**Limitations:**

No flag for negative societal impact.

Limitations have been discussed in Sec. 5.5.

---

> ### Author Rebuttal · Authors · 2023-08-10
>
> We thank Reviewer 1JJW for the appreciation of our work, positive review and the constructive discussion. We discuss some of the questions raised by the reviewer here.
>
> ### Minor Weaknesses
> > 1. **Scale Ambiguity**: ... an inherent ambiguity of any methods based on a single view but I wonder if a simple regularization to push the density of the NeRF towards the central area of the volume could have helped to standardize the scale of the fitted 3D models.
>
> We adopt a slightly different idea to achieve this goal. The density of the radiance field is initialized with a gaussian at the center with a chosen standard deviation. This biases the model towards a specific scale. The same idea is used in DreamFusion[1], RealFusion[14] and NeualLift-360 [15].
>
> > 2. **Possibly entangled text inversion**: ...  An example of unwanted entangle representation could be to pick up on the background of the single view. Randomly cutting and pasting the foreground object into random positions over random backgrounds might be a viable solution to reduce this risk.
>
> This is definitely a possible issue when performing textual inversion on a single image. Using segmented foreground with random backgrounds might help alleviate this issue to some extent. Our approach of carefully initializing the learned special token also addresses this issue to some extent.
>
> Specifically, as briefly discussed in the supplementary material, we first classify the reference image using CLIP. This can be done by finding the noun $n$ in the CLIP vocabulary that minimizes the distance between the CLIP text embedding of "A photo of <$n$>" and the CLIP image embedding. The embedding of the special textual inversion token is then be initialized with the text embedding of $n$. This ensures that the token focuses on the object in the image. For example, for an image of a red cat, the textual inversion token would be initialized with "cat". Textual inversion would then ideally update the embedding to capture "red cat". However, it is still possible but less likely that the embedding would drastically drift to capture background elements.
>
> > 3.  **Some ad hoc components per experiments**:  From the manuscript it seems that the authors used the depth loss and the warm start only for some experiments but not for all of them. ... I would suggest just presenting all results with both components switched on for all experiments.
>
> We do keep both components turned on except for the ablation experiments. This was possibly not communicated properly in the text. We will rephrase the text to make this clearer.
>
> ---
>
> ### Questions
> > How are you handling the background when training with the SDS?
>
> For every update, we randomly sample a RGB color and create a uniform background with this pixel value. Thank you for pointing this out. We will add this detail to the implementation details.

---

> > ### Comment · Reviewer_1JJW · 2023-08-14
> > **Follow up**
> >
> > Thank you for the additional details.
> > My doubt have been cleared.
> > I would suggest to the authors to add the additional details about density initialization and textual inversion in the main paper or supplementary (in case they are not there).

---

### Official Review · Reviewer_ZVcc · 2023-07-05

**Soundness:** 2 fair
**Presentation:** 3 good
**Contribution:** 3 good
**Rating:** 6
**Confidence:** 4

**Summary:**

This work proposes an approach for 3D generation from a single input image. Given an input image, a monocular depth estimate and an estimated instance mask, an image constrained radiance field is optimized following the score distillation approach from DreamFusion. They key idea is to enforce a hard constraint on the radiance field that guarantees that the input image can be reconstructed exactly whereas existing methods only encourage this with a reconstruction loss.


**Strengths:**

The paper is easy to follow and well-written. Both, qualitative and quantitative analysis indicate that the proposed method outperforms the existing baselines.
In general, inferring 3D representations from single input images is an interesting and challenging task with high relevance to the community. The idea of integrating the input image as a hard constraint in optimization is intuitive and might be interesting to the community.


**Weaknesses:**

My main concern is that the central claim of the paper, that integrating the image as a hard constraint over using a reconstruction loss makes the training more robust and that “The proposed ConRad representation significantly simplifies the process of a generating a consistent 3D model for the input image” (L.55), requires more experimental support. While the supplementary video shows a few failure cases for the baselines these results could be hand-picked. Importantly, it is also not clear if the stable training indeed results from using the proposed image conditioned radiance fields. For this, I would have liked to see an ablation study where the training pipeline is exactly the same except that once an image conditioned radiance field is used (hard constraint) and once a reconstruction loss is used (soft constraint), also ablating different strengths/weights of the reconstruction loss.

Using pearson correlation as a depth loss for monocular depth was already proposed in [1] which is not cited. Further, [1] is missing from the baseline comparison and while there is no code available, a qualitative comparison could be performed similarly to the other baselines, i.e. using results from the original paper.

[1] Deng et al, “NeRDi: Single-View NeRF Synthesis with Language-Guided Diffusion as General Image Priors”, CVPR2023


**Questions:**

Please also see Weaknesses.

L.262 mentions that NeuralLift360 uses mask supervision. However, this was not clear to me from the original NeuralLift360 paper. Could you please explain again how NeuralLift360 uses foreground masks and provide a pointer where this is stated in the original paper?

Your approach seems to achieve a higher image fidelity than the baselines. Is this only a result of more stable training or which component, e.g. compared to NeuralLift360, enables the higher image fidelity?

Minor:
L.277 “improved metrics”, it is not shown that the metrics improve over existing metrics, so I suggest removing “improved”


**Limitations:**

In general, the limitations and the broader impact were adequately discussed. From the supplementary video it looks like sometimes objects are not fully opaque (bird). This should be added to the limitations section.

---

> ### Author Rebuttal · Authors · 2023-08-10
>
> We thank Reviewer ZVcc for the constructive feedback and suggestions. We address the reviewer's comments here with additional experiments and discussion.
>
> > 1. My main concern is that the central claim of the paper, that integrating the image as a hard constraint over using a reconstruction loss makes the training more robust and that “The proposed ConRad representation significantly simplifies the process of a generating a consistent 3D model for the input image” (L.55), requires more experimental support.
>
> Thank you for raising the concern that this claim does not seem sufficiently supported. Here we will provide additional discussion and results, and re-iterate some of our intuitions for this claim.
>
> - Our proposed approach ConRad removes any need for additional reconstruction loss objectives. This in turn eliminates any associated hyperparameters. Furthermore, existing works (like RealFusion and NeuralLift-360) perform an "alternating optimization" strategy to optimize two separate objectives which is generally an unstable optimization algorithm (except in special cases). Therefore, we believe that removal of these components intuitively leads to a *simplification* of the process.
>
> - Nevertheless, we agree that additional experimental evidence could help support this claim further. Based on the reviewer's suggestion, In Table 1 of the rebuttal PDF, we compare ConRad to models constructed by learning the representation using the reconstruction losses (for RGB and foreground mask). We investigate different strengths of reconstruction losses $\lambda$ and experiment with 20 ShapeNet objects. We observe that on most objects (18 out of 20), ConRad produces better reconstructions based on the "All View $d_{oracle}$" metric. We also observe the same result across all six metrics but omit this due to space constraints.
>
>
> > 2. Using pearson correlation as a depth loss for monocular depth was already proposed in [1] which is not cited. Further, [1] is missing from the baseline comparison and while there is no code available, a qualitative comparison could be performed similarly to the other baselines, i.e. using results from the original paper.
> [1] Deng et al, “NeRDi: Single-View NeRF Synthesis with Language-Guided Diffusion as General Image Priors”, CVPR2023
>
> Thank you for pointing us to this reference. We will include this citation. To the best of our knowledge, this was unpublished work at the time of submission. However, we will add comparison to this work in the final version.
>
> > 3. L.262 mentions that NeuralLift360 uses mask supervision. However, this was not clear to me from the original NeuralLift360 paper. Could you please explain again how NeuralLift360 uses foreground masks and provide a pointer where this is stated in the original paper?
>
> This detail is not mentioned in the NeuralLift-360 paper. Please refer here for the implementation detail:
> - https://github.com/VITA-Group/NeuralLift-360#data-preparation
> - https://github.com/VITA-Group/NeuralLift-360/blob/main/nerf/utils_neurallift.py#L377-L383
> - https://github.com/VITA-Group/NeuralLift-360/blob/main/nerf/utils_neurallift.py#L620-L628
>
> > 4. Your approach seems to achieve a higher image fidelity than the baselines. Is this only a result of more stable training or which component, e.g. compared to NeuralLift360, enables the higher image fidelity?
>
> We believe that the higher fidelity can be attributed to the stability achieved by capturing the reference view accurately using ConRad. There are no other major differences compared to NeuralLift-360 and RealFusion (except the addition of depth loss).
>
> > 5. Minor: L.277 “improved metrics”, it is not shown that the metrics improve over existing metrics, so I suggest removing “improved”.
>
> We will update this in the paper.

---

> > ### Comment · Reviewer_ZVcc · 2023-08-15
> >
> > Thank you for the rebuttal and for providing the ablation study on the reconstruction loss. Since all my concerns were adequately addressed, I have updated my rating accordingly.

---

### Official Review · Reviewer_GS6t · 2023-07-07

**Soundness:** 4 excellent
**Presentation:** 4 excellent
**Contribution:** 4 excellent
**Rating:** 7
**Confidence:** 5

**Summary:**

This paper introduces Image Constrained Radiance Fields (ConRad), a novel 3D representation that constrains initial radiance fields to a reference view image without requiring training. ConRad is adept at accurately modeling the input image in one reference view and is effectively integrated with Dreamfusion-style training to convert single images to 3D shapes. The results illustrate that ConRad’s 3D reconstructions are of high quality and closely resemble the original input.

**Strengths:**

1. The paper introduces an innovative image-conditioned radiance field, ConRad, which is capable of accurately modeling a reference view without the need for training, and significantly enhances the optimization of single image-conditioned NeRFs. Despite its simplicity, ConRad produces remarkable results in converting single images to 3D.
2. The authors adeptly tackle several technical challenges by employing depth loss and a warm start strategy. ConRad’s effectiveness is convincingly demonstrated through both qualitative and quantitative results.
3. The ablation studies presented in Figure 4 are methodologically sound and illustrate the efficacy of each module proposed. Additionally, the authors provide insightful analyses.
4. The paper is well-structured, with clear and easily comprehensible presentation.

**Weaknesses:**

1. It would be beneficial for the authors to include information on the training time required for ConRad to convert a single image to 3D, and compare this with the training times of original Dreamfusion or Dreambooth3D. It is pertinent to understand if the ConRad representation contributes to accelerating the convergence speed of NeRF optimization.
2. The paper should explore the impact of varying hyperparameters within ConRad. Additionally, as Instant-NGP is utilized to represent the 3D scene, it would be valuable for the authors to investigate the influence of different NeRF representations.
3. ConRad still exhibits issues such as color saturation, akin to those found in Dreamfusion-style optimization. It is recommended that the authors consider incorporating recent advancements in Dreamfusion, such as ProlificDreamer, to further enhance ConRad’s performance.

**Questions:**

see the strengths and weaknesses.

---

> ### Author Rebuttal · Authors · 2023-08-10
>
> We thank Reviewer GS6t for their appreciation of our work, positive review and valuable suggestions to improve our paper.
>
> > 1. It would be beneficial for the authors to include information on the training time required for ConRad to convert a single image to 3D, and compare this with the training times of original Dreamfusion or Dreambooth3D.
>
> We hoped that ConRad would lead to faster convergence since the reference viewpoint is instantaneously captured. However, in practice, we did not observe convincing improvements in speed across all objects compared to the baseline (RealFusion). We keep the number of updates equal to RealFusion which works well for all objects. Both methods take approximately 20 minutes on a single A100 GPU. NeuralLift-360 uses twice the number of updates and takes 1 hour on the same hardware setup. We will add these details to the paper.
>
> > 2. explore the impact of varying hyperparameters within ConRad ... it would be valuable for the authors to investigate the influence of different NeRF representations.
>
> Thank you for the suggestion. We will report these in the final version. Switching the representation from Instant-NGP to other NeRF representations would require additional experimentation which precludes us from reporting it here due to time and resource limitations.
>
> > 3. ConRad still exhibits issues such as color saturation, akin to those found in Dreamfusion-style optimization. It is recommended that the authors consider incorporating recent advancements in Dreamfusion, such as ProlificDreamer, to further enhance ConRad’s performance.
>
> As an active area of research, there have been several recent advancements in this domain. Since ConRad improves the underlying representation, we believe that these advancements can be applied to ConRad. We thank the reviewer for the suggestion. We plan to investigate this and report any improvements in the final version.

---

> > ### Comment · Reviewer_GS6t · 2023-08-20
> >
> > The authors' rebuttal addresses my concerns. I will maintain my existing scores.

---

### Official Review · Reviewer_yDN8 · 2023-07-11

**Soundness:** 4 excellent
**Presentation:** 3 good
**Contribution:** 3 good
**Rating:** 6
**Confidence:** 4

**Summary:**

This paper presents ConRad, a new method for reconstructing 3D objects from a single RGB image. At the core of ConRad is a neural radiance field that by design satisfies the reference view constraint. That is, the representation will always render the input RGB image at the reference view. This hard constraint gets rid of the need for training in the reference view, and enables ConRad to be trained in a similar way as DreamFusion. Experiments show that ConRad can produce 3D reconstructions more faithful to the input and produce more consistent 3D models than RealFusion and NeuralLift-360.

**Strengths:**

- The idea of introducing reference-view reconstruction as a hard constraint is interesting and novel. It by design projects the image color to the 3D radiance field and gets rid of the need for training in the reference view.  This design also serves as a useful prior during training and better preserves the texture and properties of the input image. I believe this design would be interesting to the community.

- The qualitative comparison shows that ConRad performs much better than RealFusion and NeuralLift-360 in reference view reconstruction and multiview consistency. The quantitative results of ConRad is also significantly better.

- The writing is clear and easy to follow.

**Weaknesses:**

-  Most parts of the pipeline are based on previous works, such as the texture inversion, multiview SDS loss, and depth guidance, which makes the technical contribution not strong.

- At line 174, it is mentioned that $\eta$ is set to 0.1 and "the visibility depth for each pixel is a point on the ray beyond which the contribution of color is minimal (less than 10%)." I find this design and explanation not convincing and maybe wrong.
In my understanding, when $\eta = 0.1$, it means the ray from the camera to the visibility depth has only contributed 10% color, and the ray after the visibility depth will contribute the remaining 90% color, which is the opposite of the explanation in the paper. Therefore, I think it makes more sense to set $\eta$ to 0.9. Authors should clarify this and provide an ablation study on the choice of $\eta$.

- In Fig. 3, the authors should show the qualitative results of all baselines (RealFusion and NeuralList-360) for each example. There is enough horizontal space to do this. Besides, there are not enough qualitative comparisons in the current submission. Authors should provide more qualitative comparisons in the supplementary materials.

- For quantitative study, why not report the scores under the same setting as RealFusion for a fair comparison?

- From the video, it can be seen that the 3D object is semi-transparent in many cases.

- At line 237, it is mentioned that "Computation of visibility depth also does not significantly increase GPU memory consumption since we do not compute its gradients." However, since the method uses a depth loss, I believe the gradient need to be back-propagated through depth anyway. Then this argument does not make sense.

**Questions:**

Authors may respond to the weaknesses mentioned above.

**Limitations:**

Yes.

---

> ### Author Rebuttal · Authors · 2023-08-10
>
> We thank Reviewer yDN8 for the appreciation of our work, overall positive review and constructive feedback. We clarify the questions raised by the reviewer here.
>
> > 1. Line 174, ... I think it makes more sense to set $\eta$ to 0.9.
>
> This is an error in Equation (3). The equation should be
> $$ 1 - \frac{\int^t_{0} T(t) \sigma( r^{(i,j)}_p(t) )}{\int T(s) \sigma( r^{(i,j)}_p(s) ) } = \eta $$
> Thank you for pointing this out. We will fix this in the paper. We observed in experiments that the effect of $\eta$ varies with the volume and complexity of the object. However, setting it to 0.1 generally works for all objects. We will present additional visualizations in the final version with varying $\eta$.
>
> > 2. In Fig. 3, the authors should show the qualitative results of all baselines
>
> We have added more qualitative visualizations of the baselines in the attached rebuttal PDF. We observe that ConRad can consistently produce higher quality reconstructions (compare to the visualizations in the main paper). For fair comparisons to the baseline works, in the main paper, we chose to compare on the images chosen by the respective authors in Figure 3. We will add these additional visualizations to the supplementary material.
>
> > 3. why not report the scores under the same setting as RealFusion for a fair comparison?
>
> We faced several issues with the metric proposed in RealFusion:
> - First, the authors did not release an implementation of this metric. This required us to attempt to reproduce the results presented in RealFusion.
> - The metric presented in RealFusion requires conversion of the radiance field to a mesh using the marching cubes algorithm, followed by Iterated Closest Point (ICP) algorithm to match the estimated mesh to the ground truth mesh. Both these algorithms are known to be sensitive to the hyperparameters and would require manual tuning per object. This makes the metric difficult to reproduce.
>
> In contrast, we took inspiration from the metric proposed in NeuralLift-360. The "All View $d_ref$" metric presented in our work is the same as the metric used in NeuralLift-360. We also report additional metrics ($d_{all}$, $d_{oracle}) that build on top of this idea while maintaining simplicity. We will also release code for this metric along with the final version of the paper.
>
> > 4. From the video, it can be seen that the 3D object is semi-transparent in many cases.
>
> This is a common issue faced by image to 3D approaches that rely on NERF representations. Similar results can be observed in RealFusion and NeuralLift-360. We will include this in our discussion on limitations. One potential solution is to use mesh-based representations. However, this exploration would be beyond the scope of our work and would require additional research.
>
> > 5. At line 237, it is mentioned that "Computation of visibility depth also does not significantly increase GPU memory consumption since we do not compute its gradients." However, since the method uses a depth loss, I believe the gradient need to be back-propagated through depth anyway.
>
> This is referring to the computation of the "*visibility depth*" defined in Equation (3). This is not the same as the depth estimate used in the computation of the depth loss. It is true that the depth loss requires back propagation of gradients. This statement is alluding to the fact that the computation of *visibility depth* does not add additional memory consumption on top of that.

---

> > ### Comment · Reviewer_yDN8 · 2023-08-15
> > **Response to rebuttal**
> >
> > Thank you for the clarification and additional results. Most of my concerns are resolved, so I update my rating to weak accept. Please include these revisions in the revised version.

---

### Author Rebuttal · Authors · 2023-08-10

We thank all the reviewers for their time and feedback. We address individual comments to each reviewer separately.
Please find supporting material attached here as a PDF.

---

### Decision · Program_Chairs · 2023-09-21

**Decision:**

Accept (poster)

**Comment:**

The paper presents ConRad, a single-image 3D generation. All the reviewers are positive about its novelty (especially the ability to preserve the reference view without training) and the quality of the results. The authors rebuttal further clarified some of the initial concerns. The AC reads the discussions and agrees with the reviewers' assessment.